# Ergogenic Effect of BCAAs and L-Alanine Supplementation: Proof-of-Concept Study in a Murine Model of Physiological Exercise

**DOI:** 10.3390/nu12082295

**Published:** 2020-07-30

**Authors:** Paola Mantuano, Gianluca Bianchini, Ornella Cappellari, Brigida Boccanegra, Elena Conte, Francesca Sanarica, Antonietta Mele, Giulia M. Camerino, Laura Brandolini, Marcello Allegretti, Michela De Bellis, Andrea Aramini, Annamaria De Luca

**Affiliations:** 1Section of Pharmacology, Department of Pharmacy-Drug Sciences, University of Bari “Aldo Moro”, Orabona 4—Campus, 70125 Bari, Italy; paola.mantuano@uniba.it (P.M.); ornella.cappellari@uniba.it (O.C.); brigida.boccanegra@uniba.it (B.B.); elena.conte@uniba.it (E.C.); francesca.sanarica@uniba.it (F.S.); antonietta.mele@uniba.it (A.M.); giuliamaria.camerino@uniba.it (G.M.C.); michela.debellis@uniba.it (M.D.B.); 2Research & Early Development, Dompé farmaceutici S.p.A., Via Campo di Pile, s.n.c., 67100 L’Aquila, Italy; gianluca.bianchini@dompe.com (G.B.); laura.brandolini@dompe.com (L.B.); marcello.allegretti@dompe.com (M.A.)

**Keywords:** dietary supplements, branched-chain amino acids, L-Alanine, exercise, resistance to fatigue, preclinical study, murine model

## Abstract

Background: Branched-chain amino acids (BCAAs: leucine, isoleucine, valine) account for 35% of skeletal muscle essential amino acids (AAs). As such, they must be provided in the diet to support peptide synthesis and inhibit protein breakdown. Although substantial evidence has been collected about the potential usefulness of BCAAs in supporting muscle function and structure, dietary supplements containing BCAAs alone may not be effective in controlling muscle protein turnover, due to the rate-limiting bioavailability of other AAs involved in BCAAs metabolism. Methods: We aimed to evaluate the in vivo/ex vivo effects of a 4-week treatment with an oral formulation containing BCAAs alone (2:1:1) on muscle function, structure, and metabolism in a murine model of physiological exercise, which was compared to three modified formulations combining BCAAs with increasing concentrations of L-Alanine (ALA), an AA controlling BCAAs catabolism. Results: A preliminary pharmacokinetic study confirmed the ability of ALA to boost up BCAAs bioavailability. After 4 weeks, *mix 2* (BCAAs + 2ALA) had the best protective effect on mice force and fatigability, as well as on muscle morphology and metabolic indices. Conclusion: Our study corroborates the use of BCAAs + ALA to support muscle health during physiological exercise, underlining how the relative BCAAs/ALA ratio is important to control BCAAs distribution.

## 1. Introduction

Branched-chain amino acids (BCAAs: leucine, isoleucine, and valine) are three proteinogenic essential amino acids (EAAs) accounting for about 21% of total protein content in the human body [1]. Skeletal muscle tissue is our major reservoir of the BCAAs, which is provided by diet. BCAAs, especially leucine, and their metabolites act as protein building blocks, regulators of peptide synthesis, and, possibly, inhibitors of protein breakdown [2,3,4,5]; other physiological and metabolic functions have also been recently identified [6]. Leucine directly stimulates protein synthesis by activating the mammalian target of rapamycin (mTOR) signaling pathway and through the phosphorylation of translation initiation factors and ribosomal proteins [2,4,7]. The anabolic action of BCAAs is also related to the upregulation of glucose transporters and activation of insulin secretion [2,4,6]. In addition, the leucine metabolite β-hydroxy-β-methyl butyrate (HMB) exerts a parallel inhibitory effect on proteolysis by suppressing the ubiquitin–proteasome system [8], while branched-chain keto acids (BCKAs) prevent protein degradation in vitro [9]. These actions have beneficial effects on muscle mass in health and muscle-wasting conditions [3,4].

Therefore, BCAAs are recognized as potentially helpful dietary supplements to support skeletal muscle anabolism, particularly for athletes and fitness enthusiasts, as well as for elderly people to contrast age-related sarcopenia [3,10,11]. Focusing on physical activity, it is well-known that exercise promotes the oxidation of BCAAs under the control of energetic sensor AMP-activated protein kinase (AMPK) [2,12], and evidence suggests that the branched-chain alpha-keto acid dehydrogenase complex (BCKD)—at the second step of BCAAs catabolism—is activated during exercise by low ATP levels [3,13]. The first step of BCAAs oxidation takes place in the skeletal muscle, where the enzyme branched-chain amino acid aminotransferase (BCAT) is most active, instead of in the liver [3]. This gives a potentially unique advantage to BCAAs-based nutritional formulas compared with other AAs, since circulating BCAAs rapidly increase after protein intake and become readily available to extrahepatic tissues, acting as nutrient and metabolic sensors [3,14].

Independent studies in trained human subjects or exercised rodent models have shown that oral supplementation with BCAAs (or BCAAs-enriched mixtures) before and/or after exercise at variable doses (ranging from 77 mg/kg to 3 g/kg) promote protein synthesis, reduced muscle protein breakdown, as well as exercise-induced damage and fatigue [11,14,15,16,17,18]. The preferred ratio for L-Leucine, L-Isoleucine, and L-Valine administration is 2:1:1 [11,14,15,16,17,18], since BCAAs are quite safe if provided in similar proportions to those of mammal body protein [15]. Furthermore, the administration of leucine alone can ultimately lead to depletion of isoleucine and valine, since it activates the oxidation of all BCAAs [19]. Thus, the best supplementation is reasonably considered the one with all three AAs [19,20].

In contrast, other studies conducted on trained, healthy subjects have shown almost no benefits of BCAAs administration before, during, or after physical activity, on exercise performance, fatigue, or body composition [3,19,21,22,23]. It has been proposed that the supplement with the three BCAAs together, although preferred, could limit leucine effects due to the competition between these AAs for transport into muscle fibers [19]. Moreover, supplements with BCAAs alone may not effectively control muscle protein turnover, due to limited availability of other EAAs [3,11,19]. This hypothesis suggests that it would be preferable to combine BCAAs with other AAs, especially those involved in BCAAs metabolism.

In this regard, the degradation rate of BCAAs is highly responsive to the availability of two proteinogenic AAs, L-Alanine (ALA) and L-Glutamine (GLN), which are the main by-products of their metabolism [3]. One of the primary consequences of BCAAs-enriched diets is the increase in circulating levels of ALA, GLN, and BCKAs [24]. Thus, many effects of BCAAs supplementation are indeed mediated by ALA and GLN [3]. Interestingly, BCAAs are a major source of nitrogen required for ALA synthesis in the post-absorptive phase in man [3]. In a clinical study, more than 60% of circulating plasma ALA was derived from de novo synthesis after BCAAs supplementation, whereas only 40% came from endogenous protein [25]. In several preclinical studies, ALA-GLN administration to healthy or whole-body irradiated rat models controls the level of BCAAs catabolism and the amount in plasma [26]. Importantly, pre-exercise administration of ALA (alone or in combination with GLN or other natural compounds) reduced BCAAs catabolism and improved exercise performance in rats or mice subjected to progressive resistance exercise [3,27,28]. Moreover, ALA (+/− GLN) increased muscle mass and protein content in exercised rats, in parallel inducing a clear anti-inflammatory and cytoprotective effect, mediated by heat shock proteins HSP70-associated responses to muscle damage [29].

Therefore, the combination of ALA with BCAAs should help to preserve the pool of exogenously administered BCAAs, in parallel optimizing the effects on protein turnover. Dietary supplements combining different mixtures of BCAAs and ALA are commercially available and already in use by fitness enthusiasts and professionals [30]. However, to date, no solid preclinical data have been provided about their effects, or about the optimal BCAAs/ALA ratio required to maximize both direct and indirect effects of ALA, in parallel increasing BCAAs bioavailability. In a recent paper, the evaluation of the effects of a commercially available sports supplement containing BCAAs plus ALA was conducted in high-intensity endurance cycling tests [30]. The main finding was the reduction of perceived exertion rating and recovery times during performance [30].

In light of these considerations, our study primarily aims to reinforce the scientific evidence about the real effectiveness and safety of BCAAs exogenous supplementation, as a nutritional strategy to improve exercise capacity, alleviate fatigue, and protect from muscle damage in healthy subjects undergoing regular physical activity. To this aim, we evaluated the effects exerted in vivo and ex vivo by an oral formulation containing BCAAs upon muscle function, structure, and metabolism in C57BL/6J wild type (WT) mice subjected to a protocol of incremental, non-harmful exercise on a horizontal treadmill. In parallel, for the first time, we performed a head-to-head comparison of the effects exerted by three modified oral formulations containing BCAAs combined with different concentrations of ALA in the same animal model. This was done to clarify the potential boosting effect of ALA on BCAAs supplementation and the influence of the relative BCAAs/ALA ratio on the most relevant experimental readouts. This multidisciplinary assessment allowed us to obtain valuable information about the efficacy and safety of BCAAs and ALA supplementation in physiological conditions. Importantly, this may represent the first step for a future application of this supplementation in various muscle-wasting conditions, such as those occurring in inherited neuromuscular disorders or secondary cachectic, atrophic, and sarcopenic states [31,32].

## 2. Materials and Methods

All the experiments were conducted in conformity with the Italian Guidelines for Care and Use of Laboratory Animals (D.L.116/92) and with the European Directive (2010/63/UE). The study was approved by the national ethics committee for research animal welfare of the Italian Ministry of Health (authorization no. 816/2017-PR and 271/2019-PR). Most of the experimental in vivo and ex vivo procedures followed international guidelines for preclinical studies in neuromuscular diseases (http://www.treat-nmd.eu/research/preclinical/SOPs/).

### 2.1. Preliminary Pharmacokinetic Study

A preliminary pharmacokinetic (PK) evaluation of tested amino acid distribution in mouse plasma and skeletal muscle was conducted. A total of 24 (*n* = 6 mice per group), 10-week-old, male C57BL/6J wild type WT mice (Harlan, Bresso, MI, Italy) were used. All mice were subjected to health examinations and acceptance on arrival, then were housed in suitable cages (a maximum of 5 mice per cage) and acclimatized to local housing conditions for approximately 5 days. Mice were housed in a single, exclusive room, which was air-conditioned for a minimum of 15 air changes/h and constant conditions of temperature (22–24 °C), humidity (50–60%), and 12 h light/12 h dark cycle. Food (standard GLP diet, Mucedola, Settimo Milanese, MI, Italy) and water were available ad libitum. Mice were randomly assigned to the experimental groups on the basis of their body mass (ranging from 25 to 30 g). Clinical signs were regularly monitored to assess any reaction to treatments.

The night before the single-dose administration of labeled amino acids, animals were fasted; food was re-inserted in cages 3 h after amino acid supplementation. For each experimental group, doses of BCAAs were as follows: L-Leucine-13C6, 15N: 328 mg/kg; L-Isoleucine-13C6, 15N: 164 mg/kg; L-Valine-13C5, 15N: 164 mg/kg. L-Alanine was added to the doses of BCAAs indicated above, at a dose of 164 (*mix 1),* 328 (*mix 2*), and 492 (*mix 3*) mg/kg. Formulations were prepared by dissolving the amino acid mixture powder in 1.5% w/w citric acid aqueous solution. Mice were treated by a single-dose oral gavage at the administration volume of 15 mL/kg.

Blood samples (50–60 μL) were collected from the retromandibular vein at 15 min, 30 min, 1 h, 3 h, 8 h, and 24 h in heparinized centrifuge tubes (Heparin Vister 5000 U.I./mL), gently mixed and immediately placed on ice. Then, tubes were centrifuged (15 min, 3500× *g*, 4 °C), and plasma was collected and transferred to labeled tubes and frozen at −80 °C until further PK analysis. At the end of the study, mice were sacrificed by exsanguination under deep isoflurane anesthesia (EZ-B800 system provided by WPI, Worcester, MA, USA).

Gastrocnemius (GC) muscles were collected at sacrifice, washed in saline, dried on blotting paper, and homogenized in saline using CK14 tubes (Precellys 0.5 mL Soft Tissue Homogenizing Lysing Kit, Cayman Chemical, MI, USA) and Bertin homogenizer with a dilution 1/5 w/v (g/mL). Samples were centrifuged after homogenization. Human albumin was used as a surrogate to generate the calibration curve for amino acid analysis from 25 to 2000 μg/mL, whereas some of the samples were further diluted to 1:5 in the buffer. Both plasma and muscle samples were analyzed by clean-up and derivatization using the EZfaast amino acid analysis kit (Phenomenex, Castel Maggiore, BO, Italy) and analyzed by Plasma Ultra-Performance Liquid Chromatography–tandem Mass Spectrometry (UPLC Shimadzu LC-20AD equipped with ABSciex mass spectrometer API 4500Q). Plasma areas under the curves (AUCs, µg/mL·h) were calculated for the time interval 0–24 h, with BCAAs maximal concentration (C_max_, µg/mL), time for maximal concentration (T_max_, h), and mean residence time (MRT, h). In parallel, BCAAs levels in GC muscles (µg/g) were measured.

### 2.2. Animal Groups, Treatments, and Training Protocol

A total of 38, 10-week-old, male C57BL/6J WT mice (Charles River, Calco Italy) were used to perform the main study. All mice were acclimatized for about 1 week in the animal facility before starting the experimental protocol. Animals were housed in suitable cages (3–5 mice per cage), in a single room where appropriate conditions of temperature (22–24 °C), humidity (50–60%), and light/dark cycle (12 h/12 h) were constantly maintained for the entire duration of the study. After acclimatization, mice were assigned to each treatment group. Mice cohorts (*n* = 7–9) resulted in being homogeneous for body mass and forelimb grip strength values and were randomly assigned to exercise plus each treatment with standard formulation (*BCAAs*) or with each modified formulation (*mix 1*, *mix 2*, or *mix 3*). A group of *n* = 5 exercised mice was treated with vehicle (filtered tap water). An additional group of non-exercised (sed) mice (*n* = 5) was also available for the study, used as an internal control of the protocol of exercise, if and when necessary.

Once a week, each formulation was freshly prepared by dissolving the amino acid mixture powder in filtered tap water in order to obtain the desired final dose. The composition (in weight ratio) and the final doses (in mg/kg) are reported in Table 1. For BCAAs, a constant ratio of 2:1:1 in L-Leucine, L-Isoleucine, and L-Valine content was maintained in accordance with previously cited PK studies [2,3,14,15].

Water intake was carefully monitored to allow the adjustment of administered doses. This latter was calculated through considering the weekly amount of water consumed per cage, divided for the number of mice in the cage and normalized to their mean body mass. Mice were treated for 4 weeks, in parallel to the training period; the treatment started 1 day before the beginning of the training phase and lasted until the day of sacrifice for ex vivo experiments. Throughout the study, all mice were maintained on a controlled diet with a daily amount of chow of 4–5 g/mouse [33,34,35,36,37].

All mice underwent a 4-week protocol of physiological training on a motor horizontal treadmill (mod. Exer 3/6, Columbus Instruments, Columbus, OH, USA) in order to mimic the physical activity sustained by a regular, amateur runner. The protocol was adapted from those available from previous studies conducted in our laboratory and in other laboratories, both in C57BL/6J mice [38,39] and in animal models of neuromuscular disorders [33,34,36,37,40,41]. The exercise training consisted of 45 min running sessions, 5 days/week for 4 weeks on the treadmill. Each session started with a 15 min warm-up at a moderate velocity of 10 m/min, then increasing the speed of 1 m/min each min, until reaching the target velocity of 25 m/min. This maximal workload speed was maintained until the end of the exercise bout. All mice were adapted to the treadmill for ~3 weeks before the beginning of the training protocol, with exercise sessions (45 min, 5 days/week) at a low–moderate incremental velocity (5–15 m/min). The outcome of the 4-week exercise and treatment protocol was assessed on relevant in vivo and ex vivo readouts, described below in Section 2.3 and Section 2.4, respectively. To avoid introducing any bias, all experimental procedures, as well as data collection and analysis, were carried out by blinded experimenters. Figure 1 illustrates the experimental outline of the study.

### 2.3. In Vivo Monitoring and Functional Tests

All mice were non-invasively and longitudinally monitored for health and well-being throughout the total study period. None of the experimental groups showed signs of pain or distress or macroscopic alterations of vital functions. Body mass variations were regularly assessed at the start of each experimental week (Figure 1).

#### 2.3.1. Forelimb Grip Strength, Resistance to Exercise, and Isometric Torque

Forelimb grip strength was measured on a weekly basis, by means of a grip strength meter (Columbus Instruments, USA), according to a standard protocol [33,34,35,36,37,40,41]. Maximal force, absolute (expressed in kg force, KGF) and normalized to body mass (in KGF/kg), obtained from five repeated measurements per mouse, was used for data analysis [33,34,35,36,37,40,41]. At the beginning (T0) and the end (T4) of the training phase, all mice underwent an acute exercise resistance test on a treadmill at incremental speed to assess in vivo fatigability. Each mouse was let run until exhaustion, i.e., inability to re-start the running after a 20 s pause, and the total distance run (in m) up to that time was calculated and used for data analysis [33,34,35,36,37,40,41].

In vivo isometric torque produced by hind limb plantar flexor muscles (gastrocnemius, soleus, and other minor muscles) was measured in anesthetized mice at T4 by using the 1300A 3-in-1 Whole Animal Muscle Test System (Aurora Scientific Inc., Aurora, ON, Canada). Inhalation anesthesia (≈3% isoflurane in an induction chamber, then ≈2% isoflurane via nose cone for maintenance, both with 1.5 L/min O_2_) was delivered by using an anesthetic vaporizer (Harvard Apparatus Fluovac and Datex Ohmeda Isotec 4, Holliston, MA, USA) with an oxygen concentrator (LFY-1-5A, Longfei Group Co., Wenzhou, China; distributed by 2Biological Instruments, Besozzo, VA, Italy). The animal was positioned on a thermostatically controlled plate (36 °C); the right foot was placed on a pedal connected to a servomotor, forming a 90° angle with the secured hind limb. Contractions were elicited with the best stimulus intensity at increasing frequencies (200 ms trains at 1, 10, 30, 50, 80, 100, 120, 150, 180, and 200 Hz), via percutaneous electrical stimulation of the sciatic nerve through a pair of needle electrodes connected to a stimulator. Torque values were calculated with the Dynamic Muscle Analysis software (ASI DMAv5.201) and normalized to mouse body mass (N*mm^3^/kg). Normalized values were used to construct torque–frequency curves [36,41].

#### 2.3.2. Hind Limb Ultrasonography

The non-invasive ultrasound evaluation of hind limb volume and percentage of vascularization was performed in vivo at T4 by using an ultra-high frequency ultrasound biomicroscopy system (Vevo 2100; VisualSonics, Toronto, ON, Canada), which allows multiple image acquisition modes.

Each animal, properly anesthetized via inhalation as previously described, was placed in a ventral decubitus position on a thermostatically controlled table (37 °C) equipped with four copper leads, allowing for the monitoring of both heart and respiratory rate during the imaging session. Body temperature was also constantly monitored by using a rectal probe. Each hind limb of the animal was shaved, secured parallel to the body (foot at 90° with the limb), and covered by ultrasound gel. A three-dimensional (3D) volume scan of the hind limbs was acquired by translating the ultrasound probe parallel to the long axis of the hind limb. The multiple two-dimensional (2D) images were acquired at regular intervals in Power Doppler mode by using an MS250 probe at a frequency of 21 MHz, characterized by lateral and axial resolutions of 165 and 75 µm, respectively. At the end of the procedure, 3D images were reconstructed from previously collected multiple 2D frames and visualized with VisualSonics 3D software. This allowed calculating both the total volume (in mm^3^) and the percentage of vascularization (PV%) of the hind limb [35].

### 2.4. Ex vivo procedures

#### 2.4.1. Sample Collection, Processing, and Storage

At the end of T4 and of in vivo measurements, the ex vivo experimental phase started. All mice were sacrificed within 1 week, with an equal distribution of animals from each experimental group over this time window. Each mouse was anesthetized via intraperitoneal (i.p.) injection with a cocktail of ketamine (100 mg/kg) and xylazine (16 mg/kg). If required, an additional dose of ketamine alone (30 mg/kg) was injected to ensure longer and deeper sedation [33,34,35,36,41]. After the onset of anesthesia (~10 min), pilocarpine hydrochloride (1 mg/kg, Sigma-Aldrich, St. Louis, MO, USA) was injected and after 5 min, and saliva was collected from the oral cavity, transferred in a micro-centrifuge tube containing a protease inhibitor (2% PMSF, Sigma-Aldrich, USA), and kept on ice. Each saliva sample was clarified by centrifugation at 16,000× *g* for 10 min at 4 °C. Then, the supernatant was collected and stored at −80 °C until the determination of salivary immunoglobulin A (IgA) levels by enzyme-linked immunosorbent assay (ELISA).

Both right and left hind limb tibialis anterior (TA), extensor digitorum longus (EDL), quadriceps (QUAD), gastrocnemius (GC), and soleus (SOL) muscles, as well as vital organs (liver, heart, kidneys, spleen) were isolated and weighed for a gross examination of supplement toxicity and/or effects. Left TA muscle was embedded in a small amount of Tissue-Tek O.C.T. (Bio-Optica, Milan, Italy), immersed in isopentane cooled with liquid nitrogen (N_2_) for 60 s and then stored at −80 °C until further processing for histology, histochemistry, and immunofluorescence. Right TA muscle was snap frozen in N_2_ and stored at −80 °C until further processing for gene and protein expression analyses. Left QUAD muscle and the liver were snap frozen in N_2_ and stored at −80 °C until the determination of alanine transaminase (ALT) activity via a specific colorimetric assay was performed. Finally, blood was obtained by cardiac puncture with a heparinized insulin syringe and collected in heparinized tubes (Heparin Vister 5000 U.I./mL). Within 30 min after collection, platelet-poor plasma was obtained after two consequential centrifugation steps (20 min, 2000× *g*, 4 °C; 10 min, 10,000× *g*, 4 °C), and used fresh to measure creatine kinase (CK) and lactate dehydrogenase (LDH) by spectrophotometry.

#### 2.4.2. Muscle Histology, Histochemistry, and Immunofluorescence

Serial cross-sections (8 µm thick) from each frozen left TA muscle were transversally cut into a cryostat microtome set at −20 °C (HM 525 NX, Thermo Fisher Scientific, Waltham, MA, USA). Slides (Superfrost Plus, Thermo Fisher Scientific) were stained with different methods. Classical histological hematoxylin and eosin staining (H&E; Bio-Optica) was used to estimate TA muscle architecture and to calculate the area of damage and regeneration (including necrosis, inflammation, non-muscle areas, centronucleation) on the total area of muscle cross-section [33,34,35,36,37,40,41]. Histochemistry for the mitochondrial marker succinate dehydrogenase (SDH; Bio-Optica) was used to evaluate the percentage (%) of each fiber phenotype (slow oxidative, intermediate, fast non-oxidative), as well as the mean cross-sectional area (CSA, in μm^2^) for each fiber subtype [34,42]. Immunofluorescence (IF) staining for laminin was also used at this purpose (mean CSA of all fibers, in μm^2^). The antibodies used for the IF protocol were as follows: primary antibody anti-laminin rabbit, Sigma-Aldrich, dilution: 1:200; secondary antibody Alexa Fluor 488 donkey anti-rabbit, Thermo Fisher Scientific, dilution 1:500 [36]. The morphological features of the muscles were identified using digital images, acquired with a bright-field microscope (CX41, Olympus, Rozzano, Italy) and an image capture software (ImageJ, Olympus). For each muscle, the morphometric analysis was performed on at least five non-overlapping fields (10× magnification) of the total and constant transverse muscle section for all animals [33,34,35,36,37,40,41].

#### 2.4.3. Isolation of Total RNA, Reverse Transcription, and qRT-PCR for Gene Expression Analysis

For each mouse, total RNA was isolated from half of frozen right TA muscle by RNeasy Fibrous Tissue Mini Kit (Qiagen, Valencia, CA, USA; C.N.74704) and quantified by spectrophotometry (ND-1000 NanoDrop, Thermo Fisher Scientific, USA). Reverse transcription was performed as previously described [33,34,35,37,41]. qRT-PCR was performed using the Applied Biosystems Real-Time PCR 7500 Fast system (Thermo Fisher Scientific). Each reaction, carried out in triplicate, consisted in 8 ng of cDNA; 0.5 μL of TaqMan Gene Expression Assays; 5 μL of TaqMan Universal PCR master mix No AmpErase UNG (2x) (C.N. 4324018); and nuclease-free water, not DEPC220 treated (C.N. AM9930; all from Thermo Fisher Scientific), for a final volume of 10 μL. RT-TaqMan-PCR conditions were as follows: step 1: 95 °C for 20 s; step 2: 95 °C for 3 s; step 3: 60 °C for 30 s; steps 2 and 3 were repeated 40 times. Results were compared with a relative standard curve obtained from five points of 1:4 serial dilutions. The mRNA expression of genes was normalized to the mean of three housekeeping genes: ribosomal protein large P0 (RPLP0), glyceraldehyde-3-phosphate dehydrogenase (GAPDH), and eukaryotic translation elongation factor 2 (EEF2). TaqMan Hydrolysis primer and probe gene expression assays were ordered with the following assay IDs: RPLP0: Mm00725448_s1; GAPDH: Mm99999915_g1; EEF2: Mm01171434_g1; NADPH oxidase 2 (NOX2): Mm01287743_m1; interleukin 6 (IL-6): Mm00446190_m1; myogenin (MYOG): Mm00446194_m1; peroxisome proliferative-activated receptor γ coactivator 1α (PGC-1α): Mm01208835_m1; peroxisome proliferator-activated receptor γ (PPARγ): Mm01184322_m1; peroxisome proliferator-activated receptor β/δ (PPAR β/δ): Mm00803184_m1; myosin heavy chain 1 (MHC 1): Mm00600555_m1; myosin heavy chain 2A (MHC 2A): Mm00454982_m1; myosin heavy chain 2B (MHC 2B): Mm01332541_m1 [33,34,35,37,41].

#### 2.4.4. Determination of pAMPK/AMPK Ratio by Western Blot

Half of the frozen right TA muscle was homogenized in an ice-cold buffer containing 20 mM Tris-HCl (pH = 7.4 at 4 °C), 2% sodium dodecyl solfate, 5 mM ethylenediaminetetraacetic acid, 5 mM ethyleneglycoltetraacetic acid, 1 mM dithiothreitol, 100 mM NaF, 2 mM sodium vanadate, 0.5 mM phenylmethylsulfonyl fluoride, 10 mg/mL leupeptin, and 10 mg/mL pepstatin. Homogenates were centrifuged at 1500× *g* for 5 min at 4 °C. The supernatant was quantified using a Bradford protein assay kit (Bio-Rad Protein Assay Kit I5000001, Bio-Rad, Hercules, CA, USA). A total of 40 μg of proteins were separated on a 10% SDS-PAGE and transferred onto nitrocellulose membranes for 1 h at 150 mA (SemiDry transferblot; Bio-Rad). Membranes were blocked for 2 h with Tris-HCl 0.2 M, NaCl 1.5 M, pH 7.4 buffer (TBS) containing 5% non-fat dry milk and 0.5% Tween 20 (Bio-Rad), incubated overnight at 4 °C with primary antibodies at proper dilutions. After three washes with TBS containing 0.5% Tween 20 (TTBS), membranes were incubated for 1 h with secondary antibody labeled with peroxidase at proper dilutions. The following antibodies were used: AMPK primary antibody rabbit polyclonal (Cell Signaling Technology Inc., MA, USA; dilution 1:1000); phosphorylated AMPK (pAMPK) primary antibody rabbit polyclonal (Cell Signaling Technology Inc.; dilution 1:1000); secondary antibody anti-rabbit immunoglobulin G (IgG, Sigma-Aldrich, USA; dilution 1:5000). Both AMPK and pAMPK were normalized to β-actin, detected with the following antibodies: β-actin mouse monoclonal sc-47778 (Santa Cruz Biotechnology, Dallas, TX, USA; dilution 1:300); anti-mouse IgG peroxidase antibody (Bio-Rad; dilution 1:5000). At the end of each experiment, membranes were washed with TTBS, developed with a chemiluminescent substrate (Clarity Western ECL Substrate, Bio-Rad), and visualized on a Chemidoc imaging system (Bio-Rad). Densitometric analysis was performed using Image Lab software (Bio-Rad), which allows the chemiluminescence detection of each experimental protein band to obtain the absolute signal intensity. The density volume was automatically adjusted by subtracting the local background. The signal produced by AMPK and pAMPK was first normalized to the one produced by β-actin on each membrane. Then, the pAMPK/AMPK ratio (in arbitrary units, AU) was calculated [34,41].

#### 2.4.5. Determination of CK and LDH Plasma Levels

The enzymatic activity of CK and LDH in plasma samples (in U/L) was determined using specific commercially available diagnostic kits (CK NAC LR and LDH LR, SGM, Rome, Italy). Both the assays required the use of a spectrophotometer (Ultrospec 2100 Pro UV/Visible, Amersham Biosciences, Little Chalfont, United Kingdom) set to a wavelength of 340 nm at 37 °C, and were performed according to the manufacturer’s instructions [33,34,35,37,41].

#### 2.4.6. Determination of Salivary IgA Levels

Salivary IgA levels were determined using a commercial ELISA kit (Mouse IgA Ready-SET-Go! ELISA kit, eBioscience, Vienna, Austria), according to the manufacturer’s protocol [43]. Both absolute values (ng/mL) and those normalized to total protein content (ng/µg) were measured by using a microplate reader (Victor 3V, Perkin Elmer, Waltham, MA, USA) set at a wavelength of 450 nm, at room temperature.

#### 2.4.7. Determination of Hepatic and Muscular ALT Activity

ALT activity in the liver and in skeletal muscle was determined using a commercial colorimetric assay kit (Alanine Transaminase Activity Assay Kit Colorimetric/Fluorometric, Abcam, Cambridge, United Kingdom). Briefly, 50 mg of the frozen liver and left QUAD muscle tissue were washed in ice-cold phosphate-buffered saline (PBS) and resuspended in ~200 µL of ice-cold ALT Assay Buffer (supplied with the kit). Each sample was homogenized by using a Potter Elvehjem tissue homogenizer and centrifuged at top speed for 5 min at 4 °C. Then, the supernatant was collected, transferred into a clean tube, and kept on ice until the 96-well assay plate was prepared. The assay was performed according to the manufacturer’s instructions for the colorimetric method. ALT activity (mU/mL) was determined within 30 min by using the Victor 3V microplate reader set at a wavelength of 570 nm at 37 °C.

### 2.5. Statistics

All experimental data were expressed as mean ± standard error of the mean (SEM). Multiple statistical comparisons between groups (vehicle, *BCAAs*, *mix 1*, *mix 2*, *mix 3*) were performed by one-way analysis of variance (ANOVA), with Bonferroni’s *t*-test post hoc correction when the null hypothesis was rejected (*p* < 0.05). This allowed the evaluation of intra- and inter-group variability, as well as inter-group statistical comparison, while controlling the experiment-wise error rate for false positive (type I error). Unpaired Student’s *t*-test was used as a unique statistical test on certain occasions when a single comparison between two individual means was needed, i.e., when assessing differences between exercised, either treated or not, vs. sedentary mice, according to specific details in the text. All data collected follow with good approximation a normal distribution, being included in the 95% confidence interval of the mean; this generally allows for the clear identification of outliers, if any, and for the application of the statistical analyses described above. No outliers were found during the present study. Missing data in the results were then related only to overt technical issues during the experimental procedures, which led to the exclusion of those specific samples from the analysis [33,34,35,36,37,41].

The researchers were blinded to experiments, data collection, and analysis. To allow statistical comparison between standard (*BCAAs*) and modified formulations (*mix 1*, *2*, and *3*), we identified the standard, indicated as *mix 4* throughout the study, as the one containing only BCAAs just before performing statistical analysis.

## 3. Results

### 3.1. Pharmacokinetic Data

Plasma AUCs (µg/mL·h) calculated for the time interval 0–24 h are reported in Figure 2A. Interestingly, plasma exposure of BCAAs in ALA-treated groups was more than double compared to the *BCAAs* group (fold of increase: 1.9–2.5). For the other PK parameters analyzed, no modifications were found for MRT and T_max_, while C_max_ (µg/mL) of each amino acid was increased, particularly in *mix 1* and *mix 2* groups (Table 2). Indeed, compared to the *BCAAs* group, C_max_ plasma concentrations of valine were 3.2-fold higher for *mix 1* and 2.1-fold higher for *mix 2*; the same trend was observed for leucine at 2.1-fold (*mix 1*) and 1.7-fold (*mix 2*), and for isoleucine at 2.2-fold (*mix 1*) and 2.0-fold (*mix 2*).

Labeled BCAAs levels measured in GC muscle (µg/g) are reported in Figure 2B. A general trend increment in BCAAs muscle content was found for *mix 1*, *mix 2,* and *mix 3* groups vs. *BCAAs*, with *mix 2*-treated mice showing the highest levels. This increase resulted to be statistically significant for L-Valine and L-Leucine in the *mix 2* group vs. *BCAAs*.

### 3.2. In Vivo Data (4-Week Study)

Values for body mass (BM; g) and forelimb grip strength (KGF and KGF/kg) obtained by longitudinal measurements from T0 to T4 are shown in Figure 3A–C, respectively. BM was not significantly different between groups and throughout the experimental window (Figure 3A). In addition, BM values at T4 were similar to those observed in non-exercised mice (29.7 ± 1.7 g; *n* = 5). As shown in Figure 3B,C, at T0, all mice cohorts exhibited comparable values for both absolute and normalized maximal forelimb grip strength. All treated groups tended to have higher force values compared to vehicle-treated mice, particularly starting from T3, although data at this time point did not reach statistical significance. At T4, mice treated with *mix 1* or *mix 2* showed significantly higher absolute force values compared to the vehicle group (Figure 3B). For normalized force, a similar trend was observed, with *mix 2*-treated mice showing the highest value, which was significantly different vs. vehicle at T4 (Figure 3C). Interestingly, the exercise protocol itself did not modify forelimb grip strength, as shown by values at T4 for non-exercised counterparts (0.204 ± 0.014 KGF and 6.9 ± 0.24 KGF/kg; *n* = 5).

Figure 3D shows the increment in total distance run (m) obtained during the exhaustion test on a treadmill at T4 vs. T0 for each experimental group. Firstly, the protocol of exercise alone induced a slight non-significant increase in total distance run by mice compared to the basal sedentary condition at T4 (645 ± 55 vs. 548 ± 45 m, *n* = 5). Mice treated with *BCAAs* showed a modest increase in meters run, equal to +15%, in comparison to untreated mice, while mice treated with *mix 1*, *mix 2*, and *mix 3* showed an increment in meters run equal to +88%, +82%, and +122%, respectively, compared to vehicle-treated animals; none of these differences, although remarkable, were statistically significant due to inter-individual variability within each group (Figure 3D).

The protocol of exercise significantly increased plantar flexor torque (N*mm^3^/kg) production in mice, particularly at frequencies of stimulation from 50 to 200 Hz. Interestingly, mice treated with *BCAAs* had the highest torque–frequency curve, with significantly increased values compared to untreated sedentary (from 30 to 200 Hz) and exercised (from 80 to 180 Hz) mice (Figure 4). No significant change in torque was found after treating mice with each modified formulation (see Appendix A).

At T4, no remarkable differences were found in exercised mice, either treated or not, vs. sedentary ones, in terms of hind limb volume (19.9 ± 0.8 vs. 18.7 ± 1.2 mm^3^, *n* = 5) and percentage of vascularization (29.6 ± 8.8 vs. 30.3 ± 2%, *n* = 5) measured by ultrasonography (see Appendix A for 3D imaging of hind limb). As shown in Appendix A, a trend toward decrease in hind limb volume was observed in the *mix 2* group.

### 3.3. Ex Vivo Data (4-Week Study)

#### 3.3.1. Weights of Vital Organs and Muscles

The weights of main hind limb muscles (TA, EDL, QUAD, GC, SOL) and vital organs (liver, heart, kidneys, spleen), normalized to each mouse BM (mg/g), are shown in Table 3. A statistically significant increase in weight for TA and QUAD muscles from mice treated with *mix 1* or *mix 2* was found vs. vehicle or *BCAAs* groups, while only TA was significantly heavier in *mix 3*-treated mice. Moreover, a significant increase in SOL muscle weight vs. vehicle was found in *mix 2*-treated mice, while the EDL muscle was significantly heavier in *mix 1* and *mix 3* groups vs. *BCAAs*. This latter group did not show any significant variation in muscle weight compared to vehicle. GC muscle weight was reduced in the *mix 3* group vs. *BCAAs*. For internal organs, no statistically significant differences were found, apart from a decrease in liver weight observed in the *mix 1* group vs. *BCAAs* (Table 3).

#### 3.3.2. Muscle Histology Characterization and Myofiber Type Classification

Results from the histological evaluation of TA muscle architecture by means of hematoxylin and eosin (H&E) staining are reported in Figure 5A and Table 4. Figure 5A shows representative sections of H&E-stained TA muscles from each treatment group. As can be seen, no alterations in muscle architecture were induced by exercise and/or treatments. These qualitative observations were further confirmed by quantitative analysis of the total area of damage, specifically including the detection of possible inflammatory cell infiltrates and non-muscle areas (i.e., adipose and/or fibrotic tissue). As shown in Table 4, all groups exhibited a minimum, if any, presence of muscle tissue damage. The small, but significant, increase of infiltration observed in exercised mice with respect to sedentary counterparts (0.61 ± 0.05 vs. 0.31 ± 0.03%, *p* < 0.001) stayed in the range of physiological values due to the adaptation of healthy murine skeletal muscles to exercise [41].

In detail, for each experimental group, the total area of damage, including the amount of inflammatory cell infiltrates and non-muscle areas (i.e., fibrotic and/or adipose tissue), was calculated as the average percentage (%) of the total muscle area measured (taken as 100%) +/− SEM from the number of mice indicated in brackets. No statistically significant differences between groups were found by one-way ANOVA followed by Bonferroni post hoc correction.

Sample images and results from the evaluation of contractile and metabolic phenotype of myofibers measured by histochemistry for SDH in TA muscle are shown in Figure 5B–D [34]. The calculation of the number of fibers belonging to each subtype showed a similar myofiber distribution between the groups (Figure 5C), although a trend toward increase in the percentage of fast myofibers was observed in mice treated with *BCAAs* or *mix 2.* Exercised mice showed a mean cross-sectional area (CSA, in µm^2^) comparable to that of the sedentary basal condition (data not shown). The CSA of slow, intermediate, and fast fibers was significantly increased in mice treated with *BCAAs* or *mix 2* compared to the vehicle. Furthermore, a significant increase in the CSA of fast fibers was found in the *mix 2* group vs. *BCAAs* (Figure 5D).

This result was further confirmed by IF experiments for membrane protein laminin conducted in the same muscle (Appendix A), which evidenced a significant increase in mean CSA from all fiber types in the *mix 2* group compared to *BCAAs* (Appendix A).

#### 3.3.3. Protein and Gene Expression Analyses

Representative Western blots for phosphorylated (activated) pAMPK and total AMPK protein and the calculation of pAMPK/AMPK ratio are shown in Figure 6A,B, respectively. Considering the key role played by AMPK activation in muscle adaptive metabolic response to exercise [34,41], in this case, the additional group of sedentary mice was included in data analysis. Individual comparisons of exercised groups vs. the sedentary condition showed a significant increase in pAMPK/AMPK ratio in trained mice treated with either vehicle or *BCAAs* mixture. A notable increase was also observed in *mix 1* and *mix 2* mice groups, but not in the *mix 3* group. Only for *BCAAs* and *mix 2* groups we found a slight increase with respect to the vehicle (Figure 6B).

Figure 7 shows the results obtained from gene expression experiments in TA muscle. The exercise protocol induced a trend toward increase in the expression of the myokine interleukin-6 (IL-6) [44]. A clear trend to reduction of IL-6 expression was observed in mice treated with *mix 2* (−64%) or *mix 3* (−56%) compared to vehicle-treated mice, although these findings were non-significant (Figure 7). No remarkable changes in genes encoding for proteins that regulate muscle redox homeostasis (NOX2), differentiation (MYOG), and metabolism (PGC-1α and its effectors PPARγ and PPARβ/δ) were observed (Figure 7). In line with the results from the SDH staining, no significant change was observed in the MHC 1/(MHC 2A + 2B) ratio between the various groups, although a high inter-individual variability was observed within each group (Figure 7).

#### 3.3.4. Biomarkers Related to Muscle Damage, Immune Response, and Amino Acid Metabolism

Plasma levels (U/L) of CK and LDH enzymes are shown in Figure 8A,B, respectively. Both enzymes were considerably increased, although not significantly, in exercised untreated mice compared to sedentary counterparts (CK: 812 ± 378 vs. 290 ± 80 U/L, LDH: 944 ± 187 vs. 557 ± 48 U/L; *n* = 3–5). This increment was within a normal range of values for healthy, trained C57BL/6J mice [38,39]. Each formulation was able to modulate CK and LDH plasma levels by decreasing them to a different extent, with *mix 1* and *mix 2* groups showing the highest percentages of reduction vs. the vehicle (Figure 8A,B). In parallel, we observed that the treatment with *mix 1* significantly reduced both CK and LDH plasma levels with respect to *BCAAs* (Figure 8A,B), while only LDH was significantly lower in mice treated with *mix 2* (Figure 8B).

The levels of salivary IgA, absolute (ng/mL), and normalized to total protein content (ng/µg) are shown in Figure 8C,D, respectively. For completeness, total protein concentration for each group is shown in Appendix A. Absolute IgA levels in exercised mice treated with vehicle showed a non-significant trend toward increase compared to sedentary mice (3187 ± 967 vs. 2146 ± 646 ng/mL; *n* = 4–5), while no differences between the two groups were observed for normalized values (2.73 ± 0.82 vs. 2.53 ± 1.20 ng/µg; *n* = 4–5). Either absolute or normalized salivary IgA levels from all treated groups exhibited remarkable percentages of reduction with respect to vehicle-treated values; this decrease was statistically significant for the normalized values of *mix 2* vs. vehicle (Figure 8D).

Furthermore, the activity of ALT enzyme (mU/mL), determined in both liver and QUAD muscle (Appendix A) to assess the impact of ALA exogenous administration on the metabolism of the AA itself, did not undergo any major modification, showing only slight variations between the groups.

## 4. Discussion

The effective potential of BCAAs nutraceutical interventions is still controversial, mainly for the rate-limiting effect of the availability of AAs involved in BCAAs catabolism. Considering the dual mode of action by which ALA may impact the effect of BCAAs supplementation [25], different BCAAs/ALA ratios were tested in this study to evaluate the impact on bioavailability, biodistribution, and physiological parameters.

The preliminary pharmacokinetic analysis clearly demonstrated that ALA supplementation, independent of the BCAAs/ALA ratio, significantly enhanced plasma BCAAs availability. In parallel, the different BCAAs/ALA ratio critically influenced BCAAs distribution in muscle tissue. In fact, although a trend toward increase in BCAAs GC muscle concentration was observed with all mixtures, only *mix 2* and, to a lesser extent, *mix 3* showed a significant difference compared to *BCAAs* alone. Furthermore, to obtain a detailed picture of the fate of the tested amino acids, their disposition in urine and feces was also assessed. While ALA levels in urine were comparable to the vehicle group, the excretion of BCAAs in the *mix 2* group was lower (from 3.4- to 6.7-fold) compared to mice treated with *BCAAs* alone (data not shown). These results showed that the concentration of ALA is a critical determinant for BCAAs exposure and fate, providing a rationale for testing the effects of a prolonged administration of these formulations in exercised, healthy mice.

After 4 weeks, all amino acid formulations, either containing *BCAAs* only or BCAAs plus ALA (*mix 1*, *mix 2*, and *mix 3*), improved in vivo physical performance in exercise mice compared to vehicle, in terms of force and resistance to fatigue, with *mix 2* again showing the greatest efficacy.

Importantly, the observed order of potency, *mix 2* > *mix 3* > *mix* 1 > *BCAAs,* reflected the BCAAs content measured in skeletal muscle for the three mixtures, thus suggesting a direct relationship between functional performance and BCAAs tissue availability. Furthermore, since ALA is a major gluconeogenic precursor, its contribution to energy production could be, at least in part, responsible for the observed increased resistance during prolonged exercise [27,28,29]. It is worth mentioning that either BCAAs or ALA supplementation has been recently described as attenuating the production of brain serotonin, in part responsible for the onset of tryptophan-induced central fatigue during physical activity, as well as to reduce the release of serotonin, tryptophan, and other central fatigue markers in exercised rat models undergoing resistance training [3,6,45,46]. This underlines an additional mechanism accounting for the observed amelioration of in vivo performance, due to a systemic consequence of BCAAs and ALA availability.

The histology profile of TA muscle and biomarkers of muscle damage and metabolic sufferance, such as CK and LDH, were minimally affected by exercise, if any, as expected by the physiological nature of the treadmill protocol used [38,39]. Treated mice groups showed almost overlapping results, with a trend toward a further normalization of both morphology and biomarkers. In particular, a clear reduction in CK release was detected in plasma samples of treated mice, which was particularly evident in mixtures containing lower ALA concentrations. Interestingly, the most remarkable decrease of LDH plasma level was observed in plasma from mice treated with *mix 2*, suggesting a further protective action of this formulation on structural integrity and metabolism of skeletal muscle during long-lasting exercise [11,29].

In line with the above results, neither exercise nor its combination with each treatment significantly modified fiber type composition, although a trend toward increase in the percentage of fast myofibers was observed in mice treated with *mix 2* or *BCAAs*. Notably, the mean myofiber CSA was significantly higher in mice treated with *mix 2* compared to those treated with *BCAAs*; in parallel, mice treated with *mix 2* had significantly heavier hind limb muscles compared to those treated with vehicle or *BCAAs*, supporting a hypertrophic action of the combination. The impact of this effect on muscle function needs to be better understood, considering the complex role played by myofiber proportion and size in force generation [42]. However, these morphological observations could account for some of the different in vivo effects of the *mix* vs. BCAAs [47]. In parallel, we found that this group of mice showed a slight decrease in ultrasonographic hind limb volume compared to other groups, either treated or not. This apparent inconsistency could be reasonably explained by a reduction in components other than hind limb muscle mass, e.g., fat mass, which would require further investigations by dedicated experiments.

Interestingly, the training protocol significantly increased AMPK activation, according to the key role of this protein in modulating adaptation to exercise in physiological conditions, and in line with our previous observations [34,41]. In treated mice cohorts, the pAMPK/AMPK ratio was slightly fluctuating, with a trend toward increase in mice treated with *mix 2* and with *BCAAs* only. Although modest, this effect of BCAAs supplementation is consistent with their signaling roles in specific nutrient-sensing systems, and particularly for AMPK protein, which is directly involved in the fine and complex regulation of BCAAs catabolism [2,3].

In the same muscle, the expression of genes involved in skeletal muscle adaptive response to exercise (PGC-1α, PPARγ, PPARβ/δ), redox homeostasis (NOX2), and myofiber differentiation (MYOG) mainly showed mild variations in response to exercise alone or in combination with each treatment. The overall modest metabolic effect of exercise and treatments in our study is in line with the result of SDH staining for fiber phenotype, confirming the aim of the experimental approach (duration and intensity of exercise), which was not intended to induce a remarkable and clear myofiber metabolic phenotype shift, but to explore the effect of amino acid supplementation under moderate training conditions. A clear reduction in gene expression of myokine IL-6, which tended to be increased by exercise, was found in mice treated with *mix 2* and *mix 3.* This result further supports a concentration-dependent ability of ALA supplementation in attenuating inflammatory response at the tissue and systemic levels in terms of the context of exercise. This is in line with previous observations that showed that BCAAs and ALA, when individually administered, exert a potent anti-inflammatory effect in both exercised animal models and athletes [29,48], and that IL-6 release can be modulated by nutritional interventions in trained subjects, with either BCAAs or ALA reducing post-exercise levels of this cytokine by 40% [49]. The ability to control inflammatory response may also help to preserve the equilibrium of BCAAs metabolism, since inflammation and endoplasmic reticulum stress play a critical role in regulating BCAAs uptake in adipocytes and muscle tissue [50,51].

In parallel, the amino acid formulations may play a role in modulating the immune response in exercised WT mice, as assessed by measuring the levels of salivary IgA. The release of this immunoglobulin isotype in the oral cavity serves to monitor innate immune defenses and is related to training levels, and thus overtraining is associated with an “open window” of immunodepression with lower salivary IgA concentration [52]. The production of IgA in saliva may be mediated by several factors, such as stress hormones, nutritional factors, circadian cycle, hydration, alcohol intake, and also cytokines. Thus, the observed decrease of IgA levels in treated mice is coherent with the proposed effect of the supplements in regulating the inflammatory response, but could also suggest the possible induction of a mild immunosuppressive reaction, which brings IgA levels just below the normal range for this animal model [52,53].

## 5. Conclusions

The experimental findings, presented herein, support the rationale for the use of BCAAs nutritional supplements in the context of moderate and constant exercise activity in healthy subjects, as well as the potential usefulness of their combination with L-Alanine. Our head-to-head comparison showed that the formulation with the greatest overall efficacy on the main in vivo and ex vivo parameters related to exercise was *mix 2*, containing BCAAs (2:1:1) + 2ALA, in relation to increased systemic and tissue action of the AA.

In particular, it is likely true that the exogenous supplementation of ALA positively affects BCAAs bioavailability by changing the equilibrium in the BCAA–BCKA cycle, thereby decreasing BCAAs catabolism. Interestingly, our data also demonstrate that the BCAAs/ALA ratio strongly influences BCAAs distribution in muscle tissue, thus providing a plausible rationale to the specific effects observed with *mix 2* in our animal model.

Further experiments will be needed to assess the long-term effects of this formulation to support muscle health in more sustained exercise regimens or in light of a future application as a potentially useful nutritional supplement to counteract muscle-wasting, which characterizes many pathological conditions in children and adults.

## 6. Patents

The compounds used for this study were provided by Dompé farmaceutici S.p.A. with patent application no. 102019000010401.

## Figures and Tables

**Figure 1 nutrients-12-02295-f001:**
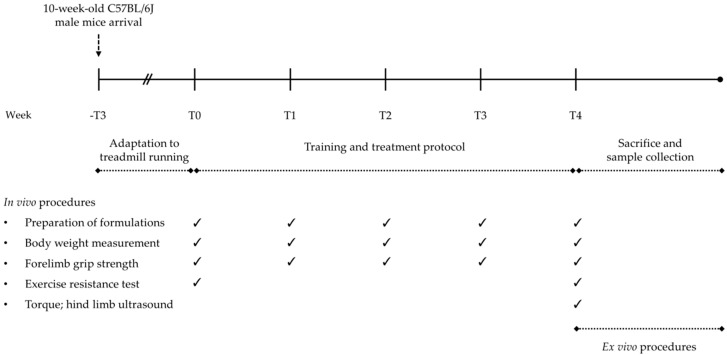
Scheme illustrating experimental design and timeline of the study.

**Figure 2 nutrients-12-02295-f002:**
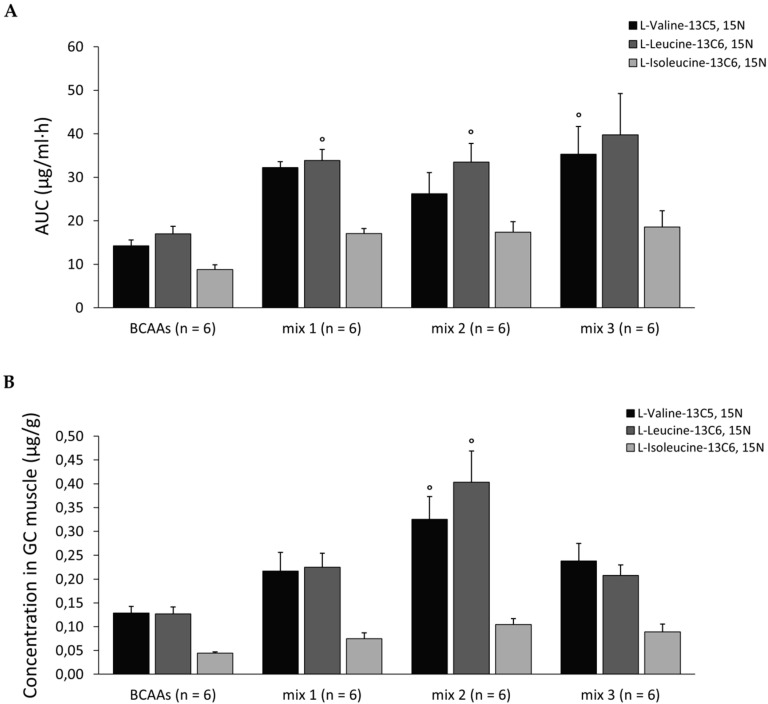
(**A**) Plasma area under the curves (AUCs) of each labeled amino acid calculated for the time interval 0–24 h for mice treated with branched-chain amino acids (*BCAAs*), *mix 1*, *mix 2*, or *mix 3*. (**B**) Concentration for each labeled amino acid measured in gastrocnemius (GC) muscle for all groups. Values are expressed as mean ± standard error of the mean (SEM) from the number of mice indicated in brackets. One-way ANOVA followed by Bonferroni post hoc correction was performed to compare all groups for each amino acid. A statistically significant difference among groups was found by ANOVA for both (**A**) (*F* > 4.9, *p* < 0.0001) and (**B**) (*F* > 5.1, *p* < 0.001). Bonferroni post hoc for individual differences between groups is as follows: ° vs. *BCAAs* (*p* < 0.05).

**Figure 3 nutrients-12-02295-f003:**
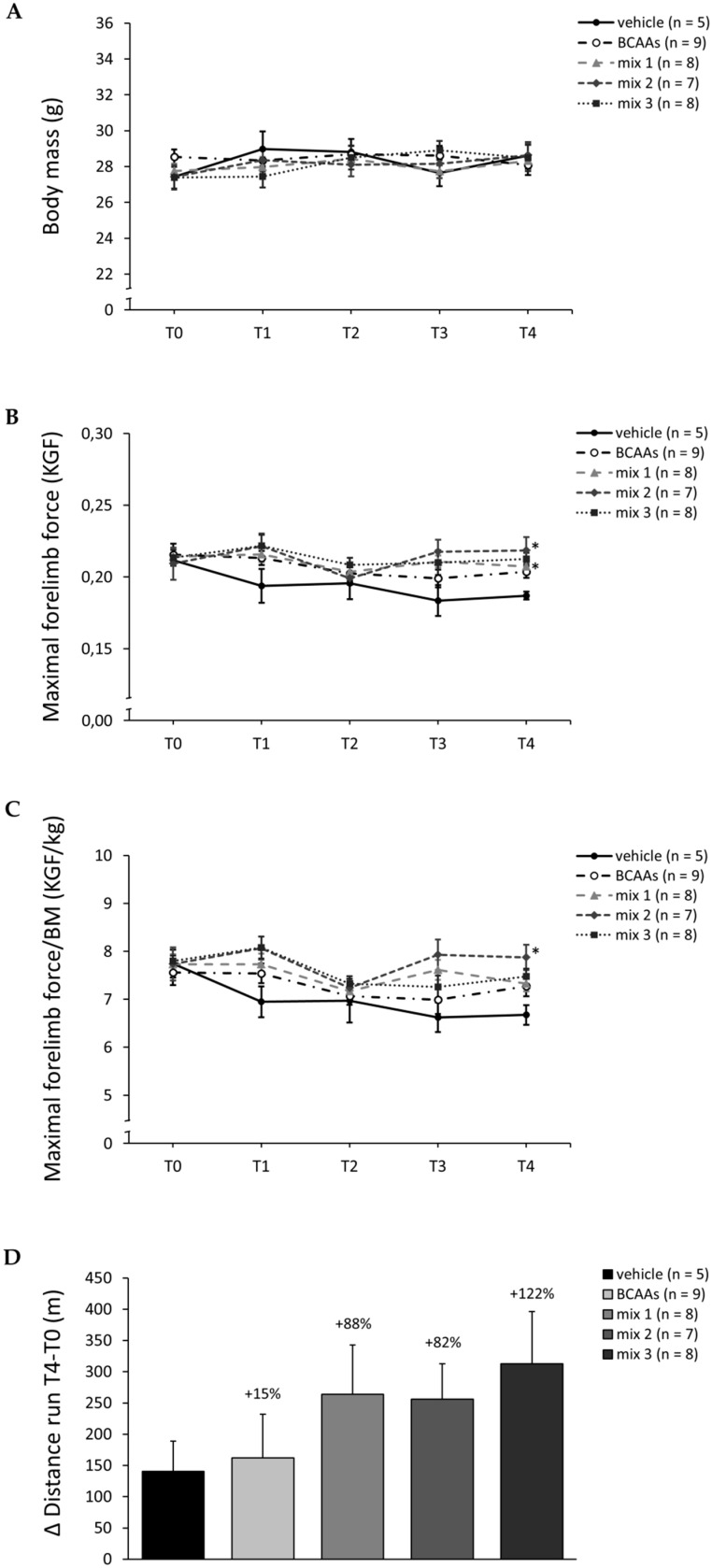
(**A**–**C**) Variations in body mass (BM g; (**A**)) and maximal forelimb grip strength, both absolute (KGF; (**B**)) and normalized to BM (KGF/kg; (**C**)), at time points T0, T1, T2, T3 and T4 for exercised mice treated with vehicle, *BCAAs*, *mix 1*, *mix 2*, or *mix 3*. Values are expressed as mean ± SEM from the number of mice indicated in brackets. One-way ANOVA followed by Bonferroni post hoc correction was performed to compare all groups. For (**A**), no statistically significant differences were found at any time point. For (**B**,**C**), a statistically significant difference among groups was found by one-way ANOVA at T3 (*F* > 4.1, *p* < 0.03) and T4 (6.5, *p* < 0.008). Individual differences between groups were found only at T4 by Bonferroni post hoc, as follows: * vs. vehicle (0.001 < *p* < 0.005). (**D**) Increment in distance run (m) on treadmill during the exercise resistance test performed at T4 with respect to values obtained at T0 for mice from each experimental group. Values are expressed as mean ± SEM from the number of mice indicated in brackets. No statistically significant differences between groups were found. For each treated group, the percentage increase with respect to vehicle is indicated above the bars.

**Figure 4 nutrients-12-02295-f004:**
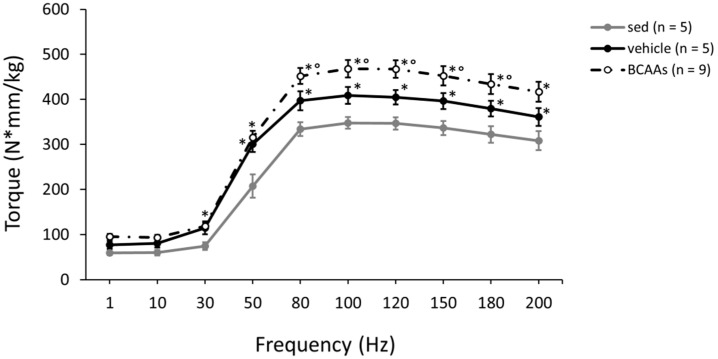
The graph shows the values of plantar flexor torque (N*mm/kg) produced at increasing stimulation frequencies (from 1 to 200 Hz), obtained from sedentary (sed) mice and exercised mice treated with vehicle or *BCAAs* at T4. Values are expressed as mean ± SEM from the number of mice indicated in brackets. At frequencies from 30 to 200 Hz, a statistically significant difference among all groups was found by one-way ANOVA (*F* > 5.4, *p* < 0.02). Bonferroni post hoc for individual differences between groups is as follows: * vs. sed (6.2 × 10^−11^ < *p* < 0.02), ° vs. vehicle (*p* < 0.004).

**Figure 5 nutrients-12-02295-f005:**
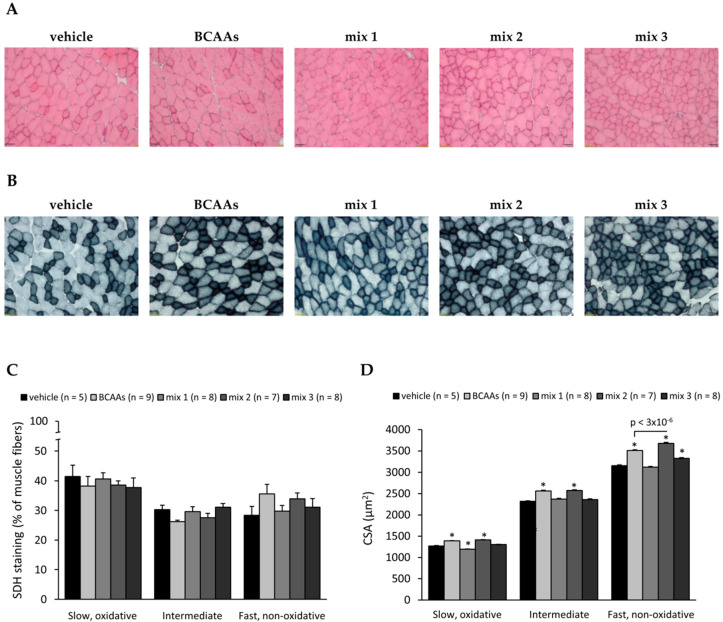
(**A**) Representative TA muscle sections stained with hematoxylin and eosin (10× magnification) from exercised mice treated with vehicle, *BCAAs*, *mix 1*, *mix 2*, or *mix 3*. All muscles exhibited the typically well-organized architecture of a healthy murine muscle, with no signs of damage, abnormal inflammatory infiltrates, or fibrotic areas. (**B**) Representative TA muscle sections stained for succinate dehydrogenase (SDH) histochemistry (10× magnification) for each group. This staining allowed us distinguish between oxidative (darker) and less oxidative/non-oxidative (lighter) myofibers in each section due to the different levels of SDH activity. (**C**) Mean percentage (%) of each myofiber phenotype (slow, intermediate, and fast) with respect to the total number of myofibers (taken as 100%) +/− SEM obtained from the number of mice per group indicated in brackets. (**D**) Cross-sectional area (CSA, µm^2^) for each myofiber phenotype, expressed as mean ± SEM from the number of mice per group indicated in brackets. One-way ANOVA followed by Bonferroni post hoc correction was performed to compare all groups. In (**D**), for each fiber type, statistically significant differences among groups were found by one-way ANOVA (slow: *F* > 51; *p* < 2.5 × 10^−8^; intermediate: *F* > 46; *p* < 1.5 × 10^−7^; fast: *F* > 73; *p* < 1.2 × 10^−9^). Bonferroni post hoc for individual differences between groups is as follows: * vs. vehicle (1.6 × 10^−11^ < *p* < 5.7 × 10^−5^). The statistical difference found by Bonferroni post hoc for *mix 2* vs. *BCAAs* is indicated above the bars. No significant differences were found for *mix 1* or *3* vs. *BCAAs*.

**Figure 6 nutrients-12-02295-f006:**
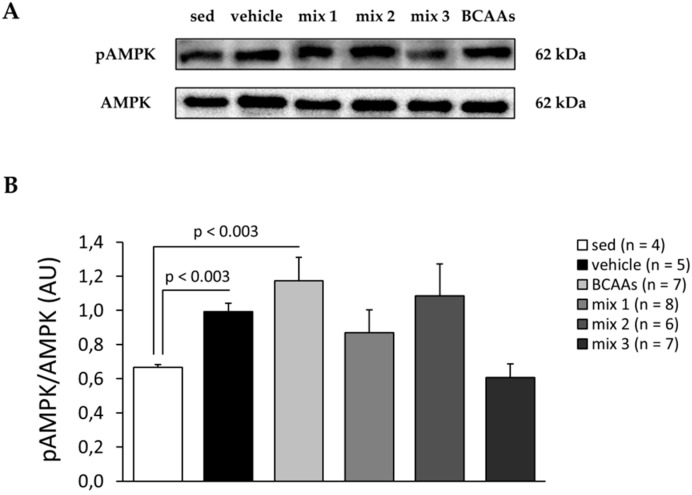
(**A**) Representative Western blots of phosphorylated AMP-activated protein kinase (pAMPK) and total AMPK performed in TA muscles from sedentary mice and exercised mice treated with vehicle, *BCAAs*, *mix 1*, *mix 2*, or *mix 3*. (**B**) Calculation of the pAMPK/AMPK ratio for each experimental group. Values are expressed as mean ± SEM from the number of mice indicated in brackets. No statistically significant differences between vehicle and treated groups were found by one-way ANOVA followed by Bonferroni post hoc correction. Individual comparisons of the means vs. sedentary mice showed statistically significant differences found by unpaired Student’s *t*-test, which are indicated above the bars.

**Figure 7 nutrients-12-02295-f007:**
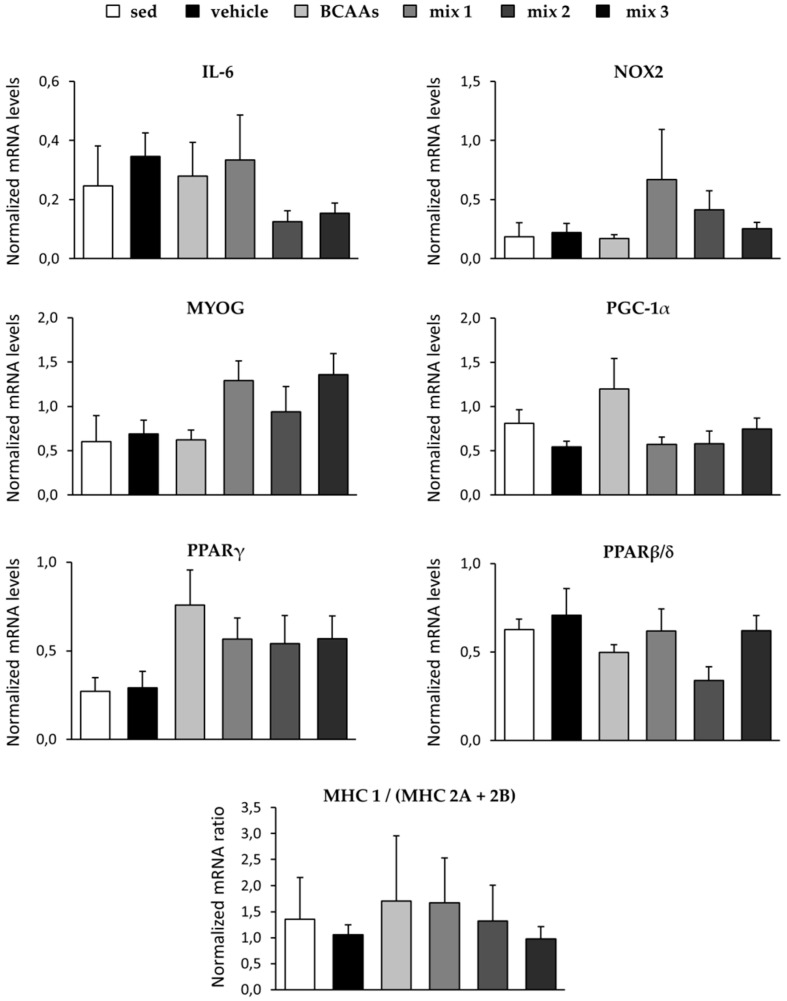
The figure shows the transcriptional levels, measured by qRT-PCR in TA muscle, of genes encoding for proteins involved in skeletal muscle adaptive response to exercise (interleukin 6, IL-6; peroxisome proliferative-activated receptor γ coactivator 1α, PGC-1α; peroxisome proliferator-activated receptor γ, PPARγ, and PPARβ/δ), redox homeostasis (NADPH oxidase 2, NOX2), and myofiber differentiation (myogenin, MYOG), as well as of myosin heavy chain (MHC) 1, 2A, and 2B genes, presented as MHC 1/(MHC 2A + 2B) ratio, for sedentary mice and exercised mice treated with vehicle, *BCAAs*, *mix 1*, *mix 2*, or *mix 3*. Values are expressed as mean ± SEM from 4–8 samples per group, normalized to the mean expression of three selected muscle housekeeping genes (ribosomal protein large P0, RPLP0; glyceraldehyde-3-phosphate dehydrogenase, GAPDH; eukaryotic translation elongation factor 2, EEF2). No statistically significant differences between vehicle and treated groups were found by one-way ANOVA followed by Bonferroni post hoc correction.

**Figure 8 nutrients-12-02295-f008:**
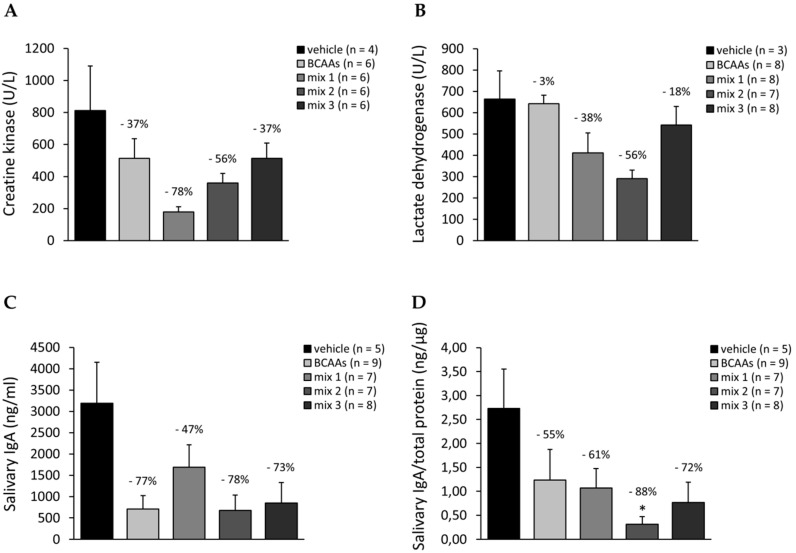
(**A**,**B**) Levels of enzymes creatine kinase (U/L, (**A**)) and lactate dehydrogenase (U/L, (**B**)), measured in plasma samples of exercised mice treated with vehicle, *BCAAs*, *mix 1*, *mix 2*, or *mix 3*. Values are expressed as mean ± SEM from the number of mice indicated in brackets. No statistically significant differences between vehicle and treated groups were found by one-way ANOVA followed by Bonferroni post hoc correction. The values of percentage reduction vs. the vehicle group are indicated above the bars. (**C**,**D**) Salivary IgA levels expressed as absolute (ng/mL, (**C**)) and normalized to total protein (ng/µg, (**D**)) for each cohort. Values are expressed as mean ± SEM from the number of mice indicated in brackets. One-way ANOVA followed by Bonferroni post hoc correction was performed to compare all groups. Statistically significant differences among groups were found by one-way ANOVA for both (**C**) (*F* > 3.9; *p* < 0.04) and (**D**) (*F* > 6; *p* < 0.01). For (**D**), individual differences between groups were found by Bonferroni post hoc, as follows: * vs. vehicle (*p* < 0.004). The values of percentage reduction vs. the vehicle group are indicated above the bars.

**Table 1 nutrients-12-02295-t001:** The composition and the final dose of each tested formulation.

Formulation	Composition: BCAAs + ALA (Weight Ratio of L-Leu: L-Ile: L-Val: L-Ala)	Final Dose (mg/kg)
*BCAAs*	2:1:1	656
*mix 1*	2:1:1:1	820
*mix 2*	2:1:1:2	984
*mix 3*	2:1:1:3	1148

**Table 2 nutrients-12-02295-t002:** The table shows the C_max_ (µg/mL) of labeled amino acids for mice treated with *BCAAs*, *mix 1*, *mix 2,* or *mix 3*.

PK Parameters—C_max_ (µg/mL)			
Group	L-Valine	L-Leucine	L-Isoleucine
*BCAAs* (*n* = 6)	6.54 ± 0.62	11.14 ± 3.52	5.18 ± 1.59
*mix 1* (*n* = 6)	21.21 ± 2.77 °	23.04 ± 3.19	11.19 ± 1.58
*mix 2* (*n* = 6)	13.88 ± 2.94	18.82 ± 2.92	10.37 ± 2.03
*mix 3* (*n* = 6)	13.15 ± 2.13	10.93 ± 2.11	5.61 ± 0.96

Values are expressed as mean ± SEM from the number of mice per group indicated in brackets. One-way ANOVA followed by Bonferroni post hoc correction was performed to compare all groups for each amino acid. A statistically significant difference among groups was found by ANOVA for valine (*F* > 4.7; *p* < 0.0001). Bonferroni post hoc for individual differences between groups is as follows: ° vs. *BCAAs* (*p* < 0.05).

**Table 3 nutrients-12-02295-t003:** The table shows the weight of hind limb tibialis anterior (TA), extensor digitorum longus (EDL), quadriceps (QUAD), gastrocnemius (GC), and soleus (SOL) muscles, and the weight of organs (liver, heart, kidneys, spleen), normalized to mice body mass (BM; mg/g).

Group	Weight of Hind Limb Muscles/BM (mg/g)					Weight of Vital Organs/BM (mg/g)			
	TA	GC	EDL	QUAD	SOL	Liver	Heart	Kidneys	Spleen
vehicle (*n* = 5)	1.7 ± 0.02	2.2 ± 0.02	0.38 ± 0.03	6.1 ± 0.57	0.25 ± 0.01	53.2 ± 3.4	5.2 ± 0.10	5.9 ± 0.11	2.8 ± 0.15
*BCAAs* (*n* = 9)	1.6 ± 0.03	2.3 ± 0.03	0.32 ± 0.01	6.9 ± 0.10	0.30 ± 0.02	57.8 ± 2.3	5.4 ± 0.12	5.9 ± 0.17	3.3 ± 0.17
*mix 1* (*n* = 8)	2.1 ± 0.03 *^,^°	2.1 ± 0.03	0.42 ± 0.02 °	7.8 ± 0.18 *^,^°	0.36 ± 0.01	42.6 ± 2.3 °	5.6 ± 0.33	6.5 ± 0.12	3.3 ± 0.24
*mix 2* (*n* = 7)	1.9 ± 0.02 *^,^°	2.4 ± 0.12	0.37 ± 0.02	7.9 ± 0.17 *^,^°	0.47 ± 0.04 *	47.2 ± 2.9	5.9 ± 0.27	6.4 ± 0.21	3.9 ± 0.63
*mix 3* (*n* = 8)	1.8 ± 0.03 °	2.1 ± 0.02 °	0.44 ± 0.03 °	7.1 ± 0.18	0.32 ± 0.03	52.1 ± 3.3	5.4 ± 0.19	6.3 ± 0.14	3.7 ± 0.36

Values are expressed as mean ± SEM from the number of mice per group indicated in brackets. One-way ANOVA followed by Bonferroni post hoc correction was performed to compare all groups. Statistically significant differences among groups were found by one-way ANOVA for both muscles (*F* > 4.2; *p* < 0.03) and organs (*F* > 3.2; *p* < 0.04). Bonferroni post hoc for individual differences between groups is as follows: * vehicle (8.4 × 10^−9^ < *p* < 0.001); ° vs. *BCAAs* (9.7 × 10^−12^ < *p* < 0.004).

**Table 4 nutrients-12-02295-t004:** The table shows the quantitative evaluation of TA muscle histology performed by hematoxylin and eosin (H&E) staining, performed on at least five non-overlapping fields (10× magnification) of a transverse muscle section.

TA Muscle Histology (H&E Staining)			
Group	Total Area of Damage (%)	Infiltration (%)	Non-Muscle Area (%)
vehicle (*n* = 5)	0.62 ± 0.05	0.61 ± 0.05	0.015 ± 0.012
*BCAAs* (*n* = 9)	0.48 ± 0.11	0.44 ± 0.11	0.035 ± 0.019
*mix 1* (*n* = 8)	0.79 ± 0.15	0.76 ± 0.15	0.023 ± 0.017
*mix 2* (*n* = 7)	0.52 ± 0.11	0.50 ± 0.10	0.023 ± 0.015
*mix 3* (*n* = 8)	0.48 ± 0.09	0.48 ± 0.09	0.00

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
