# Peer review of "Ergogenic Effect of BCAAs and L-Alanine Supplementation: Proof-of-Concept Study in a Murine Model of Physiological Exercise"

_nutrients, 2020, doi:10.3390/nu12082295_

Round 1

Reviewer 1 Report

In 20 years of reviewing manuscripts, this is best 'first' draft I have had the pleasure to referee. The rationale for the study is well presented and defended; the methods are appropriate (laboratory specific coefficients of variation for the various dependent variables - particularly blood assays - would have been nice to present but not necessary), and conclusions reasonable from the presented results. My only critique is that it reads a little long, but am typically not in favor of shortening articles as long as they stay under journal requirements.

I do not even have a simple "copy edit," but I would say that "physiological exercise" does not need to be a keyword. 

Congratulations on this fine paper.

Author Response

Answer

We sincerely thank the Reviewer for appreciating our study and our manuscript and we are grateful for such positive comments.

We also agree with the Reviewer about the observation regarding text length; thus, some paragraphs have been shortened in order to make it more readable, without affecting the content of the manuscript. “Physiological exercise” has been removed, as suggested, and replaced by “Exercise” in the keywords list.

Reviewer 2 Report

This manuscript aims to examine the effects of BCAAs in combination with varying concentrations of L-alanine on physical performance, muscle morphology and histology, pharmacokinetic parameters, protein and gene expression, biomarkers, and weights several organs in mice. Overall, this is an interesting study with valuable findings, however there are some major concerns that must be addressed.

Major:

-In general, the manuscript is too long and must be condensed. There are several instances where redundant or unnecessary information is included and should be removed or condensed to make the manuscript easier to follow. Specifically, the Results section is 11 pages long plus the supplementary files. Please do not include any methodological procedures in the Results, as these should only be included in the Methods. Edit the Methods to include what variables are measured in each procedure to refrain from including these in the Results. Also, while the Introduction and Discussion provide interesting information, much of it is unnecessary and can be condensed. The manuscript is very difficult to read as is.

-Much of what is presented in the Introduction and Discussion is not referenced. Please include more references throughout the Introduction and Discussion.

-Consider reporting the SD instead of SEM for all measures, as this is what is typically reported in statistical analyses. This will also make it easier for other researchers to compare their results to those in the current study.

-Please add a figure outlining the experimental design. There are many variables, groups, and timepoints examined in this study, making it difficult for the reader to follow which procedures were done when.

-Please include specific statistical findings in the text of the Results, rather than the table/figure captions.

-Please include details for normality testing, missing data, etc. in the Statistics section.

-I am confused by the Statistical analyses run in this paper. You state: "If necessary, additional single comparisons between two means were performed by unpaired Student's t-test". Considering that you are comparing two means in a post hoc when a significant effect in an ANOVA is observed, why do you need to run t-tests? If you are comparing all groups to one another, you cannot pick and choose which groups you want to analyze using a t-test. Also, please include more details about the ANOVA (groups compared) in the Statistics section.

-The Figure and Table legends are confusing, partially because of the point above. Again, why are certain comparisons using t-tests and others aren't? I would avoid using t-tests altogether. When you report "Statistically significant differences were found by ANOVA followed by Bonferroni Post hoc", you need to include which groups are significantly different from one another. This is unclear in the figures.

-Why were some analyses completed with a vehicle, some analyses were completed with sedentary mice, and some analyses were only compared with the BCAA group? Please explain.

Minor:

-Please indicate how frequently the formulations were administered to animals.

-Please ensure that you are consistent with how you are reporting units. For example, ensure that the "L" in "mL" is capitalized consistently throughout the paper or kept in lowercase. Additionally, make sure spacing is consistent throughout the paper (ex: include space between "1h" if you are including a space between "15 min").

-Line 208: Please revise to make sure appropriate paragraphs are referenced.

-Section 2.3.1: Why are the grip strength procedures and treadmill fatigue test included in the same section if isometric torque is included in its own section? Either group all physical performance outcomes into a single section or separate all.

-Why were multiple mice assigned to one cage? This may influence outcome variables, as you are assuming that all mice received equal amounts of the supplement.

Author Response

Response to Reviewers – nutrients-861838 

Reviewer #2

Comments and Suggestions for Authors

This manuscript aims to examine the effects of BCAAs in combination with varying concentrations of L-alanine on physical performance, muscle morphology and histology, pharmacokinetic parameters, protein and gene expression, biomarkers, and weights several organs in mice. Overall, this is an interesting study with valuable findings, however there are some major concerns that must be addressed.

Answer

We thank the Reviewer for the positive comments about the interest of our manuscript and results. We also thank for the detailed criticisms raised to improve the manuscript. Our point-by-point response and the description of the changes made in the main text are detailed below. We hope that both answers and revisions clarify all the issues brought up by the Reviewer.

Major:

- In general, the manuscript is too long and must be condensed. There are several instances where redundant or unnecessary information is included and should be removed or condensed to make the manuscript easier to follow. Specifically, the Results section is 11 pages long plus the supplementary files. Please do not include any methodological procedures in the Results, as these should only be included in the Methods. Edit the Methods to include what variables are measured in each procedure to refrain from including these in the Results. Also, while the Introduction and Discussion provide interesting information, much of it is unnecessary and can be condensed. The manuscript is very difficult to read as is.

Ans.: We understand the Reviewer’s concerns about manuscript length and readability. We carefully revised the text, with particular attention to the points raised herein. We do believe that the new version and the revised parts are indeed clearer and easier to follow.

- Much of what is presented in the Introduction and Discussion is not referenced. Please include more references throughout the Introduction and Discussion.

Ans.: We apologize if reference citing was incomplete or insufficient. Both Introduction and Discussion have been carefully revised to address this point.

- Consider reporting the SD instead of SEM for all measures, as this is what is typically reported in statistical analyses. This will also make it easier for other researchers to compare their results to those in the current study.

Ans.: Although we understand the Reviewer’s concern of SD being typically reported in statistical analysis, we do believe that, in fact, both SEM and SD are commonly used in biomedical research studies, depending on the case and mostly on the nature of the parameter(s) and its measurement. In this particular setting, and based on our previous studies, we think that the use of SEM instead of SD would be preferable. SD is, in fact, a pure measure of deviation of experimental data from theoretical ones, due to the intrinsic nature of the set of data itself (e.g. mouse body weight in a cohort of age- and sex-matched animals), and this is independent from the sample size. As opposite, SEM takes into account both intrinsic variations within the data set and the potential errors of the measure (e.g. due to the weight scale, to the operator, etc.), and it is in fact dependent on the sample size and the number of repetitions: the larger is the size of the sample data, the smaller is the SEM. This, in fact, corrects the potential bias introduced by the measure. In light of these considerations, we have extensively used SEM in place of SD when describing parameters that have a normal distribution. By the way, SEM contains SD (which can be simply calculated) and is therefore suitable for proper statistical analysis and comparison with data from previous experiments from the same laboratory and from other labs (see, for instance, Capogrosso et al., FASEB J 2018). In addition, ANOVA has been used to compare groups, then intrinsic variance of the data is properly taken into account.

- Please add a figure outlining the experimental design. There are many variables, groups, and time points examined in this study, making it difficult for the reader to follow which procedures were done when.

Ans.: We thank the Reviewer for this suggestion. A figure (Fig. 1) reporting the experimental plan and design has been added to the revised version of the manuscript. We agree that this increases the clarity of the plan for the readers.

- Please include specific statistical findings in the text of the Results, rather than the table/figure captions.

Ans.: We appreciate the Reviewer’s advice and we remodulated the Results. However, due to the complexity of our study, we believe that it is more immediate that a description of specific statistical findings is included in tables/figures captions to make the head-to-head comparison between groups easier to follow.

- Please include details for normality testing, missing data, etc. in the Statistics section.

Ans.: The requested details have been provided in the Statistics paragraph.

- I am confused by the Statistical analyses run in this paper. You state: "If necessary, additional single comparisons between two means were performed by unpaired Student's t-test". Considering that you are comparing two means in a post hoc when a significant effect in an ANOVA is observed, why do you need to run t-tests? If you are comparing all groups to one another, you cannot pick and choose which groups you want to analyze using a t-test. Also, please include more details about the ANOVA (groups compared) in the Statistics section.

Ans.: We apologize for possible confusion in the text about the statistical analysis. We fully agree with the Reviewer’s point of view about the usefulness of ANOVA plus Bonferroni post hoc.  We  carefully revised both the presentation of statistical analysis, figures, tables (with relative captions) and main text, trying to avoid any misunderstanding or un-necessary repetition of statistical tests. We better explained which groups have been analyzed with ANOVA and for which purposes. We specified the experimental data for univocal use of the unpaired Student’s t-test.

- The Figure and Table legends are confusing, partially because of the point above. Again, why are certain comparisons using t-tests and others aren't? I would avoid using t-tests altogether. When you report "Statistically significant differences were found by ANOVA followed by Bonferroni Post hoc", you need to include which groups are significantly different from one another. This is unclear in the figures.

Ans.: Again, we apologize if the information regarding the statistical analysis resulted unclear. Please, refer to our previous answer for a main general response. We carefully revised the manuscript tried to avoid any confusion or unclear statement in text, figure, tables and legends about statistical analysis.

- Why were some analyses completed with a vehicle, some analyses were completed with sedentary mice, and some analyses were only compared with the BCAA group? Please explain.

Ans.: All the analyses presented in the manuscript have been performed in the main five experimental groups, including exercised mice treated with vehicle, BCAAs, mix 1, mix 2 or mix 3. This, to allow evaluate the effects of each formulation, either standard (BCAAs) or modified (mix 1, mix 2 or mix 3) with respect to untreated mice, as well as a direct comparison of the effects of each mixture vs BCAAs alone. In addition, as described in the Methods section, sedentary mice have been included in the study as an internal control of the training protocol outcome, in order to confirm its non-harmful nature. This was evidenced by the lack of differences found by comparing exercised and non-exercised treated mice for most of the parameters analyzed herein, whose results have been always reported in the main text for completeness. When considered relevant, i.e. for in vivo torque and gene and protein expression, sedentary mice have been included in the main figures. For the PK study, only groups treated with labeled (and then, detectable) amino acids, were analyzed.

Minor:

- Please indicate how frequently the formulations were administered to animals.

Ans.: The frequency of administration has been indicated as follows:

Lines 135 – 136: “Mice were treated by a single-dose oral gavage at the administration volume of 15 ml/kg”.

Lines 167 – 168: “Once a week, each formulation has been freshly prepared by dissolving the amino acid mixture powder in filtered tap water, to obtain the desired final dose.”.

- Please ensure that you are consistent with how you are reporting units. For example, ensure that the "L" in "mL" is capitalized consistently throughout the paper or kept in lowercase. Additionally, make sure spacing is consistent throughout the paper (ex: include space between "1h" if you are including a space between "15 min").

Ans.: This has been corrected throughout the text.

- Line 208: Please revise to make sure appropriate paragraphs are referenced.

Ans.: We apologize for the typing error, appropriate paragraphs have been indicated.

- Section 2.3.1: Why are the grip strength procedures and treadmill fatigue test included in the same section if isometric torque is included in its own section? Either group all physical performance outcomes into a single section or separate all.

Ans.: We thank the Reviewer for the suggestion, torque has been included in the Section 2.3.1.

- Why were multiple mice assigned to one cage? This may influence outcome variables, as you are assuming that all mice received equal amounts of the supplement.

Ans.: The choice of assigning multiple mice per cage has been based on our long and positive experience in preclinical studies in which nutraceutical and pharmacological treatments were chronically administered to adult mice in drinking water [e.g. Capogrosso et al., Pharmacol Res 2016; Capogrosso et al., FASEB J 2018; Mantuano et al., Biochem Pharmacol 2018; Mele et al., Transl Res 2019]. As described in detail in the Methods section, the water intake was constantly monitored to allow the adjustment of administered doses. Mean water consumption per mouse per week (resulting in ~ 4,5 ml water mouse/day) was calculated based on the amount of water consumed per cage, divided by the number of animals in the cage and normalized to their mean body weight, these latter indeed showing no significant variations for each cohort throughout the protocol (as reported in Figure 3A). Here and in previous works, this method allowed us to estimate in a quite accurate manner the mean dose assumed by each treated mouse, keeping mice socially housed as recommended by international guidelines for laboratory animal care, since isolation is associated with depressive states that could affect treatment outcome evaluation [Kalliokoski et al., PLoS ONE 2014]. Some of this aspects have been briefly included in the revised version.

Round 2

Reviewer 2 Report

Thank you for making several of the suggested edits. While I feel that the paper has been improved, I believe there are numerous concerns that still need to be addressed.

-The introduction still needs an extensive review of citations and references. It does not look like any additional references were added into the introduction from the last revision.

-Please change "body weight" throughout the manuscript to "body mass"

-I am still confused by the statistical analysis. It seems that only certain variables were analyzed by t-tests, regardless of whether or not there was a significant ANOVA value. For example, Figure 7 states that there were no statistically significant differences between the vehicle and treated groups when analyzed by ANOVA, however you still choose to use a t-test to examine a similar question (and found significant differences from the t-test). This does not seem like best statistical practice. I would try to avoid using t-tests altogether or use the in every analysis, rather than picking and choosing which variables to run t-tests on. The same issue is present for Figure 8.

-Results for Pharmacokinetic data: Why was this not analyzed as an ANOVA and only as t-tests?

-Line 167: Change "has"

-Lines 410-411: Please remove the sentence "From T1 to T4, all treated groups resulted to be stronger than the vehicle-treated one," as this data did not reach statistical significance so there is no difference between groups. Alternatively, include a statement about the data being non-significant.

-Line 436: Did the post hoc reveal no significant differences at T3, since there are no markers indicated in the graphs at T3.

-Line 525: You write "the percentage increase and the statistical difference..... are indicated above the bars," however there is no percentage increase included in these figures.

-The first 2 paragraphs of the discussion can be condensed or removed entirely since this is already discussed in the introduction.

-Line 681: Please rephrase.

Author Response

Round 2 – Reviewer #2

Comments and Suggestions for Authors

Thank you for making several of the suggested edits. While I feel that the paper has been improved, I believe there are numerous concerns that still need to be addressed.

Answer

We thank the Reviewer for appreciating our efforts made to improve the manuscript based on the suggested edits. We hope that this second revised version will address all further issues raised by the Reviewer.

Our point-by-point response and the description of the changes made in the main text are detailed below.

-The introduction still needs an extensive review of citations and references. It does not look like any additional references were added into the introduction from the last revision.

Ans.: As suggested, new references have been added to the Introduction in the revised manuscript version.

-Please change "body weight" throughout the manuscript to "body mass"

Ans.: We understand the concern of the Reviewer. The required change has been made throughout the main text, captions and figures.

-I am still confused by the statistical analysis. It seems that only certain variables were analyzed by t-tests, regardless of whether or not there was a significant ANOVA value. For example, Figure 7 states that there were no statistically significant differences between the vehicle and treated groups when analyzed by ANOVA, however you still choose to use a t-test to examine a similar question (and found significant differences from the t-test). This does not seem like best statistical practice. I would try to avoid using t-tests altogether or use the in every analysis, rather than picking and choosing which variables to run t-tests on. The same issue is present for Figure 8.

Ans.: We are sorry for the still present unclarity about statistics. There was indeed no picking and choosing about variables to run t-tests on, and we surely would not like this to be perceived by the reader. To avoid any confusion or misunderstanding the results of the Student’s t test analysis, in the few cases still present, have been deleted, with particular attention to Figure 7 and Figure 8.

In the new revised version, the only situation in which Student’s t-test is still present, refers to individual comparisons vs sedentary mice. The above changes have been better detailed in the proper section.

-Results for Pharmacokinetic data: Why was this not analyzed as an ANOVA and only as t-tests?

Ans.: Student’s t-test was chosen as preferential simple test considering the acute nature of PK experiments, that needs to assume a simple model, in consideration of the intrinsic inter-individual variability and the complexity of the dynamic and multi-phase PK process. However, we again understand the Reviewer’s concern and we wish to avoid any ambiguity. Therefore, ANOVA followed by Bonferroni post hoc has been applied also in this case. The necessary changes have been made in the main text (paragraph 3.1), Figure 2 A,B and Table 2.

-Line 167: Change "has"

Ans.: This has been corrected in the main text.

-Lines 410-411: Please remove the sentence "From T1 to T4, all treated groups resulted to be stronger than the vehicle-treated one," as this data did not reach statistical significance so there is no difference between groups. Alternatively, include a statement about the data being non-significant.

Ans.: Thank you for pointing this out. The phrase has been reworded using an adequate statement.

-Line 436: Did the post hoc reveal no significant differences at T3, since there are no markers indicated in the graphs at T3.

Ans.: Yes, ANOVA resulted to be significant at T3 (F > 4.1, p < 0.03), but no individual differences between groups were revealed by Bonferroni post hoc. This has been better detailed in Figure 3 caption.

-Line 525: You write "the percentage increase and the statistical difference..... are indicated above the bars," however there is no percentage increase included in these figures.

Ans.: We apologize for the oversight. This has been corrected.

-The first 2 paragraphs of the discussion can be condensed or removed entirely since this is already discussed in the introduction.

Ans.: We agree with the Reviewer about this point. The first two paragraphs of the Discussion sections have been further shortened and redundancies with respect to the Introduction have been removed.

-Line 681: Please rephrase.

Ans.: Thank you for the suggestion. The sentence has been rephrased.
